# Optical properties of meteoric smoke analogues

*Tasha Aylett[1], James S. A. Brooke[1], Alexander D. James[1], Mario Nachbar[2], Denis Duft[2], Thomas Leisner[2,3], John M. C. Plane[1]*

(1) School of Chemistry, University of Leeds, Leeds, UK.

(2) Institute for Meteorology and Climate Research, Karlsruhe Institute of Technology (KIT), Karlsruhe, Germany

(3) Institute of Environmental Physics (IUP), Ruprecht-Karls-University Heidelberg, Heidelberg, Germany

**Abstract**

Accurate determination of the optical properties of analogues for meteoric smoke particles (MSPs), which are thought to be composed of iron-rich oxides or silicates, is important for their observation and characterization in the atmosphere. In this study, a photochemical aerosol flow reactor (PAFS) has been used to measure the optical extinction of iron oxide MSP analogues in the wavelength range 325-675 nm. The particles were made photochemically, and agglomerate into fractal-like particles with sizes on the order of 100 nm. Analysis using Transmission Electron Microscopy (TEM), Energy Dispersive X-ray spectroscopy (EDX) and Electron Energy Loss Spectroscopy (EELS) suggested the particles were most likely maghemite-like ($\gamma$-$Fe_2O_3$) in composition, though a magnetite-like composition could not be completely ruled out. Assuming a maghemite-like composition, the optical extinction coefficients measured using the PAFS were combined with maghemite absorption coefficients measured using a complementary

experimental system (the MICE-TRAPS) to derive complex refractive indices that reproduce
both the measured absorption and extinction.

## 1. Introduction

The ablation of cosmic material in the mesosphere leads to the formation of nanometer-sized
meteoric smoke particles (MSPs) (Plane et al., 2015). Reaction, condensation and subsequent
agglomeration of stable reservoir species such as FeOH, Mg(OH)$_2$, NaHCO$_3$ and SiO$_2$ leads to
the formation of MSPs over a timescale of several days. However, very little is known about the
physical and chemical properties of these particles. It is important to establish the composition
and other characteristics of MSPs because the particles are thought to be involved in a wide
range of atmospheric processes as they are transported down through the atmosphere, including:
mesospheric metal chemistry; mesospheric oxygen chemistry; nucleation of polar mesospheric
clouds (PMCs); stratospheric aerosol chemistry (including the nucleation of polar stratospheric
clouds (PSCs)); and deposition of bioavailable metal sulfates into the oceans (Plane et al., 2015).
The detection and characterisation of MSPs has proven extremely challenging as the
mesosphere-lower thermosphere (MLT) is a notoriously difficult region in which to perform in
situ studies. In terms of the composition, at present only two types of investigations exist: rocket-
borne instruments (e.g. Faraday cup detectors and electric work-function studies) and remote
sensing (e.g. optical spectroscopy). The only direct measurements have been obtained *via*
sounding rocket flights, though only charged particles have been sampled with any success. One
example is the ECOMA (Existence and Charge state Of Meteoric smoke particles in the middle
Atmosphere) project (Rapp et al., 2010). This work constrained the MSP size and work function,
with electronic structure calculations inferring a likely MSP composition of Fe and Mg
hydroxide clusters with low silica content (Rapp et al., 2012).
Important progress has also been achieved using remote sensing techniques: the SOFIE (Solar
Occultation for Ice Experiment) instrument on the AIM (Aeronomy of Ice in the Mesosphere)
satellite has detected MSPs by optical extinction, conducting solar occultation measurements
from April 2007 to the present. Extinction measurements at 330, 867 and 1037 nm were used to
show that the best-fit particle compositions are iron-rich oxides (magnetite (Fe$_3$O$_4$), wüstite
(FeO), magnesiowüstite (Mg$_x$Fe$_{1-x}$O, $x$=0 -0.6)) or iron-rich olivine (Mg$_{2x}$Fe$_{2-2x}$SiO$_4$, $x$=0.4-0.5)
(Hervig et al., 2017). That is, the major meteoric elements Fe, Mg and Si are either mixed in
olivinic particles with a single average composition, or MSPs are a mix of metal oxide and silica
particles. However, this technique makes an important assumption: that the bulk (crystalline)
refractive indices (RIs) used to infer smoke compositions are applicable to MSPs, despite
evidence that the particles are structurally amorphous, fractal-like agglomerates (Saunders and
Plane, 2006). This assumption is currently not confirmed, and as such it is important to measure
RIs of realistic MSP analogues, especially those of iron-rich particles.
A number of crystalline MSP analogues (Fe$_2$O$_3$, silica (SiO$_2$) and iron silicates (Fe$_x$Si$_{(1-x)}$O$_3$ ($0 \le$
$x \le 1$))) with radii on the order of 2 nm have recently been generated in the laboratory using a
low pressure, non-thermal microwave resonator (Nachbar et al., 2018a;Nachbar et al., 2018c).
The particles are transferred to a low pressure, supersaturated particle trap - the Molecular Flow
Ice Cell/Trapped Reactive Atmospheric Particle Spectrometer (MICE/TRAPS) - in which
particle properties can be determined. The latest work using this system derived absorption
efficiencies for $Fe_2O_3$ particles at 450, 488 and 660 nm (Nachbar et al., 2018c). David et al.
(2012) have demonstrated the production of maghemite particles with a similar experimental
arrangement. In fact, Navrotsky et al. (2008) have argued that maghemite is thermodynamically
favored with respect to hematite for particles smaller than 16 nm in diameter. The particles
produced in the study of Nachbar et al. (2018b) are therefore very likely to have been
maghemite.
Amorphous MSP analogues have previously been generated in the laboratory using a
photochemical aerosol flow system (PAFS) (Saunders and Plane, 2011, 2010, 2006). Particles
with compositions close to the minerals hematite ($\alpha$-$Fe_2O_3$), goethite (FeOOH), fayalite
($Fe_2SiO_4$) and silica ($SiO_2$) were produced when metal-containing precursors were photolysed in
the presence of $O_3/O_2$. Particle size distributions were measured using a Scanning Mobility
Particle Sizer (SMPS), and optical extinction measurements were obtained for comparison with
values calculated from Mie theory using literature RIs for the unidentified particles. Although the
experimental size distributions of the MSP analogues produced could be replicated using an
agglomeration model (Jacobson, 2005;Saunders and Plane, 2010, 2006), there was significant
uncertainty in the measured size distribution. Consequently, Mie theory was able to reproduce
the measured extinction using bulk RIs for $\alpha$-$Fe_2O_3$ and $Fe_2SiO_4$ particles, though not when using
the experimental size distribution.
In this manuscript, the photochemical technique used by Saunders and Plane has been developed
further to study the agglomeration and optical properties of iron oxide particles. The measured
optical extinction has been modelled using Mie theory and the Rayleigh-Debye-Gans (RDG)
approximation (discussed below). Analysis using Transmission electron microscopy (TEM),
Electron Energy Loss Spectroscopy (EELS) and EDX (Energy Dispersive Xray) spectroscopy
indicate a maghemite-like ($\gamma$-$Fe_2O_3$) particle composition. The measured optical extinction data
was combined with absorption efficiencies from Nachbar et al. (2018c) to derive wavelength-
dependent complex RIs that reproduce the measured extinction. Maghemite particles have not
been previously considered in the compositional analysis of MSPs because no RIs are available
in the literature. However, the formation of maghemite nanoparticles in laboratory studies *via*
two different production methods (*via* photolysis/in microwave plasma) that operate under
distinctly different conditions demonstrates the potential relevance of this species in the
atmosphere, and the need for further studies on this compound as a potential candidate for MSPs.

**2. Experimental Methods**
**2.1 PAFS**
The photochemical apparatus used to generate analogue MSPs (Figure 1) has been described
previously (Saunders and Plane, 2006, 2010, 2011). The setup consists of a cylindrical glass
photolysis cell with quartz end windows ($r$ = 4 cm; $\phi$ = 25 cm) into which a combined flow of
the Fe precursor, iron pentacarbonyl vapour ($Fe(CO)_5$), and $O_3/O_2$ was introduced. The $Fe(CO)_5$
was generated by passing a flow of $N_2$ through a round-bottomed flask containing ~3 cm$^3$ of
liquid $Fe(CO)_5$ (Aldrich) cooled in a water-ice bath to 0 °C. The round-bottomed flask and ice-
bath were covered to prevent any premature photolysis (and subsequent build-up of material on
the flow tube walls). $O_3$ was produced by photolysing $O_2$ at 184 nm, by passing a flow of $O_2$
through a glass cell with a quartz window in front of a Hg pen lamp. Once in the photolysis cell,
the gases were irradiated using a 1000 W ozone-free Xenon arc lamp. Variable $N_2$ 'curtain'
flows were passed across each of the cell windows such that the total flow rate was 550 sccm (1
sccm = 1 cm³ min⁻¹ at standard temperature and pressure (273 K and 1 bar)). After leaving the
photolysis cell the particle flow was directed through an absorption cell ($r = 10$ cm; $\phi = 48$ cm)
with White cell optics in which the optical extinction of the particles was measured (hereafter
referred to as the White cell). On exit from the White cell, particle size distributions were
recorded using a SMPS consisting of a differential mobility analyser (DMA) and a condensation
particle counter (CPC). The DMA sheath and aerosol flow rates were 3 L min⁻¹ and 0.3 L min⁻¹
respectively, with a scan taken every 3 minutes (a scan time of 120 seconds and retrace of 30 s
was used).

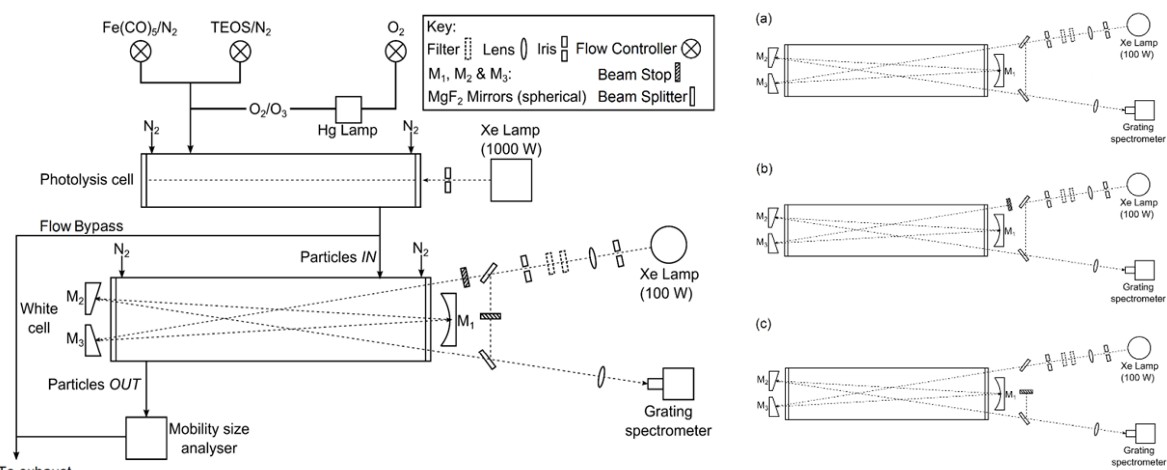

Figure 1. Schematic diagram of the experimental system used for the generation and optical detection of
MSP analogues, where a, b & c show different beam configurations used to generate a normalised cell
spectrum.
Light from a 100W Xenon arc lamp was focused into the cell using a quartz lens (focal length =
75 cm). The lamp intensity could be controlled by the insertion of a selection of neutral density
filters, and stray lamp light was eliminated using an iris. Borosilicate White cell windows
prevented further photolysis by the spectroscopy lamp, and excluded second-order light at
wavelengths below ~330 nm from entering the spectrometer and potentially contaminating the
spectra. $N_2$ curtain flows (500 sccm) were passed across the windows to prevent aerosol
deposition. Particle extinction was measured between 325 and 675 nm.  A total optical path
length ($l$) of 624 cm was achieved by folding the light path 12 times between three concave
mirrors comprising the White cell. The exit beam was focused with a quartz lens onto a fibre
optic coupled to an Acton Research Spectra Pro 500i spectrometer, in which the light was
dispersed using a grating (150 groove mm⁻¹) onto a CCD camera. The exposure time was 0.1 s,
with 57 accumulations per spectrum generating 9 spectra per minute.
To measure smaller levels of optical extinction than in our previous work (Saunders and Plane,
2006, 2010, 2011), an optical by-pass was introduced to normalize for drifts in the Xe lamp
spectral intensity with time. Beam stops were used to create three different beam configurations
from which the separate signals could be extracted (Figure 1). Spectra were recorded in three

139 minute cycles with one minute of spectra taken using each configuration: (a) the beam directed
140 through the cell and additionally through the bypass ($I_a$); (b) the beam directed only through the
141 bypass ($I_b$); and (c) the beam directed only through the cell ($I_c$). This generated one normalized
142 spectrum ($I_n$) every three minutes (see equation E1). The minimum detectable absorbance ranged
143 from 0.07 to 0.004 over the wavelength range studied.

144 E1 $$I_n(\lambda) = \big(I_a(\lambda) - I_b(\lambda)\big)/\big(I_a(\lambda) - I_c(\lambda)\big),$$

145 In a typical experiment, the sample flows were switched on, with the particle flow initially
146 diverted to an exhaust rather than through the White cell. The background particle size
147 distribution and optical intensity in the White cell ($I_{n,bg}(\lambda, t)$) were then measured for about 30
148 minutes. The particle flow was then directed through the White cell, and a further 21 minutes of
149 sample measurements were recorded ($I_{n,sa}(\lambda, t)$). Thereafter, the particle flow was diverted back
150 to the exhaust and background measurements resumed for around 45 minutes. A repeat sample
151 measurement was recorded followed by approximately 30 minutes of reference measurements
152 until the peak of the recorded size distribution had stabilized to within 1 %.

153 The gas-phase spectrum for the $Fe(CO)_5$ precursor was measured with a PerkinElmer Lamda 90
154 UV/Vis spectrometer in a $1 \times 1$ cm gas cuvette. Due to the wide range covered by the absorption
155 cross section in the measured wavelength range (4 orders of magnitude), the final spectrum was a
156 composite of two spectra; the low-wavelength end of the spectrum ($\lambda < 280$ nm) was an average
157 of three low pressure measurements (P ~ 2 torr) and the high wavelength portion ($\lambda > 280$ nm)
158 was an average of two higher pressure measurements (P ~ 20-30 torr). A reference spectrum for
159 the empty cuvette was subtracted from each individual spectrum before averaging.

160

161 **2.2 TEM**

162 Particles formed in the photochemical aerosol flow system were collected by diverting the flow
163 bypass through a round-bottomed flask containing a suspended transmission electron microscopy
164 (TEM) grid (copper mesh with a holey carbon film coating). The grids were then stored under
165 vacuum in the dark prior to imaging. Particles were analysed using TEM with energy dispersive
166 X-ray spectroscopy (EDX) and electron energy loss spectroscopy (EELS) at the University of
167 Leeds (FEI Titan3 Themis 300).

168

169 **2.3 MICE/TRAPS**

170 Absorption efficiencies determined in Nachbar et al. (2018c) for maghemite particles with the
171 MICE/TRAPS apparatus were used in combination with the optical extinction measured in this
172 work for iron oxide particles produced with the PAFS apparatus, in order to derive complex RIs.
173 The experimental and analytical methods used for the MICE/TRAPS experiment have been
174 described in detail previously (Meinen et al., 2010a; Meinen et al., 2010b; Duft et al., 2015;
175 Nachbar et al., 2016), with the recent methodology for particle production outlined in Nachbar et
176 al. (2018a). The analysis procedure for the determination of absorption efficiencies is specified
177 in Nachbar et al. (2018c).

178 In brief, singly charged, spherical and compact nanoparticles are produced by mixing vapour
179 from a volatile precursor (solid ferrocene, $Fe(C_5H_5)_2$, ~353 K) with a flow of oxygen and helium.
180 This mixture then flows through a low pressure, non-thermal microwave resonator to create a

plasma in which metastable exited Fe is oxidised to produce $Fe_2O_3$ particles. A portion of the
flow passes into a vacuum chamber through an aerodynamic lens, a flow-limiting orifice and an
octupole ion guide (Figure S1, supplementary information). Particles of a chosen size are
deflected with a quadrupole deflector and are subsequently trapped into a cloud of ~ 1 mm radius
within the ion trap MICE, where a He bath gas is added to thermalize the particles. Within the
MICE, the particles are subject to a well calibrated concentration of gas phase $H_2O$ molecules
(Nachbar et al., 2018b). Small numbers of particles are extracted from the trap at regular time
intervals to a Time-of-Flight (ToF) mass spectrometer for particle mass determination.
In a typical experimental run, $Fe_2O_3$ particles were admitted into the MICE, where $H_2O$
molecules were adsorbed onto the particle surfaces with increasing trapping time until an
equilibrium of adsorbing and desorbing molecules was reached. A number of repeat runs were
performed where the cloud of particles was irradiated using optically pumped continuous wave
semiconductor lasers (OBIS LX, Coherent, at 405 nm, 488 nm and 660 nm), increasing the laser
power in each subsequent run. Absorption of the laser light by the particles caused heating and
desorption of $H_2O$ molecules from the particle surface (see Figure S2, supplementary
information). Parameters such as the initial mass and radius ($r$) of the particles, and the
temperature change due to irradiation could then be calculated from the mass of the levitated
nanoparticles as a function of the residence time in MICE. Assuming an equilibrium between
radiative heating and collisional cooling enables the absorption cross section ($C_{abs}$), and
absorption efficiency ($Q_{abs}$, see equation E2) to be calculated. The latter is typically used when
comparing the absorption of different sized particles.
E2
$$Q_{abs} = \frac{C_{abs}}{\pi \cdot r^2}$$


## 3. Results and Discussion

### 3.1 TEM

Examination of particles collected from the PAFS using TEM show non-spherical, fractal-like
particles. A range of particle sizes can be observed, ranging from tens of nanometers to microns
in radius (Figure 2). The fractal-like agglomerates are formed of primary spheres, whose size
was estimated by taking a number of measurements from three high resolution images of
different agglomerates, one of which is shown in Figure 2 (right-hand panel). In each of these
three images, 15 primary spheres were measured from around the visible 'edge' of the particle,
where a defined spherical shape could be seen. This analysis resulted in a primary particle radius
of 1.65 ± 0.15 nm.

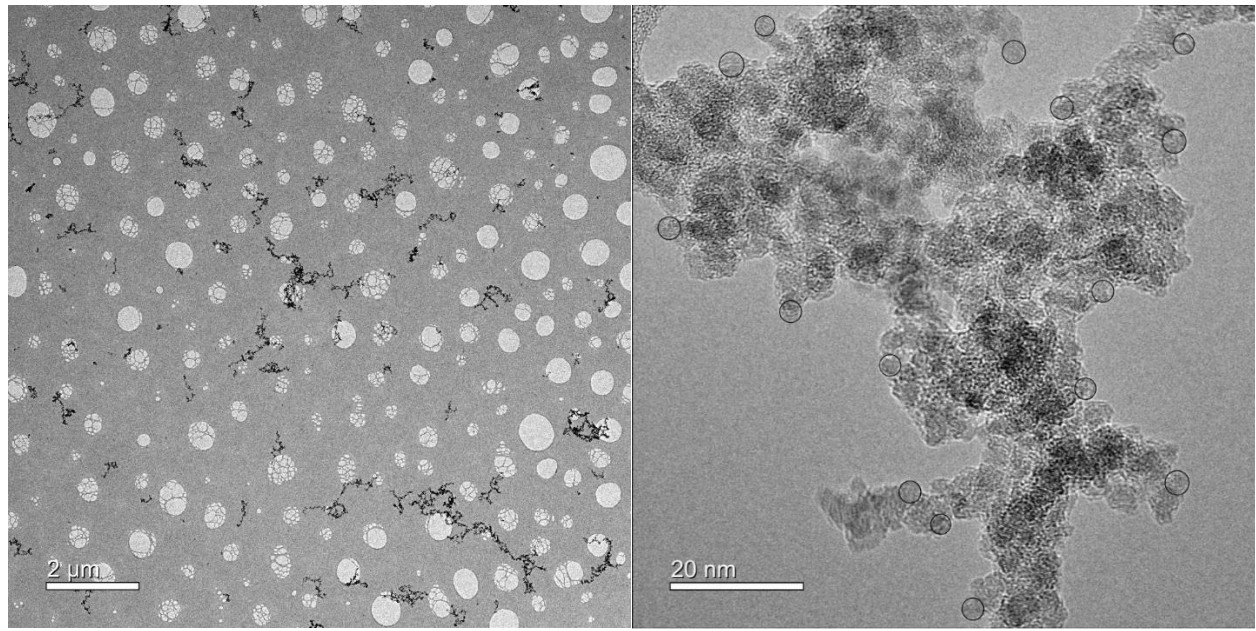


Figure 2. (Left panel) Low resolution TEM image showing the size-range of iron oxide agglomerates (dark
grey/black particles) collected on a holey-carbon grid (light grey holes and webbing). (Right panel) High
resolution TEM image showing the primary particles forming an agglomerate. Black circles indicate
measured primary spheres used for size characterisation.

219

Comparison of the background-subtracted, low-loss deconvolved Fe L-edge and O K-edge EELS
spectra with those from iron oxide standards can provide information on the particle composition
(Figure 3, left-hand panels) (Brown et al., 2017; Brown et al., 2001). The O K-edge spectra for
the iron oxide standards have been aligned using the energy loss for the peak designated as *b* in
Figure 3, due to the invariance of this peak in the spectra. Likewise, the Fe L-edge spectra have
been aligned to the sample peak *a*. On inspection of the O K-edge, a wüstite-like sample
composition can be excluded due to the differing edge-onset energy and shape of peak *a*. A
hematite-like sample composition can also be rejected on the basis of the lack of the double-peak
structure characteristic of hematite in the sample spectrum. This is corroborated on inspection of
the Fe L-edge, where a well-defined shoulder on the low energy side and a broad shoulder on the
high energy side of peak *a* are observed for hematite and wüstite, respectively, neither of which
are present in the sample spectrum.

In the case of both magnetite and maghemite there are no distinctive features in either the O K or
Fe L-edges to easily distinguish between the two species. Nonetheless, upon closer inspection
(Figure 3, right-hand panels) the sample spectra more closely resemble those for the maghemite
standard as compared to those for magnetite. For the Fe L-edge, although the shoulder on the low
energy side of peak *a* is larger in the sample spectrum than that observed for maghemite, it is
more well defined than the shoulder seen in the magnetite spectrum. Furthermore, the profile of
peak *a* more closely follows that for the maghemite standard on both the high and low energy
sides. Though a defined double-peak structure is not observed in peak *b*, both the peak profile on
the high energy side and the height of the peak more closely resemble maghemite. For the O K-
edge, although there are some differences between the sample spectra and those for both
standards (notably the lack of a defined peak *c*), there are minimal differences between the

spectra for the maghemite and magnetite standards. For this reason, though the profile of peak *a* more closely follows that of maghemite, it is not possible to distinguish between the two species from the O K-edge spectra alone.

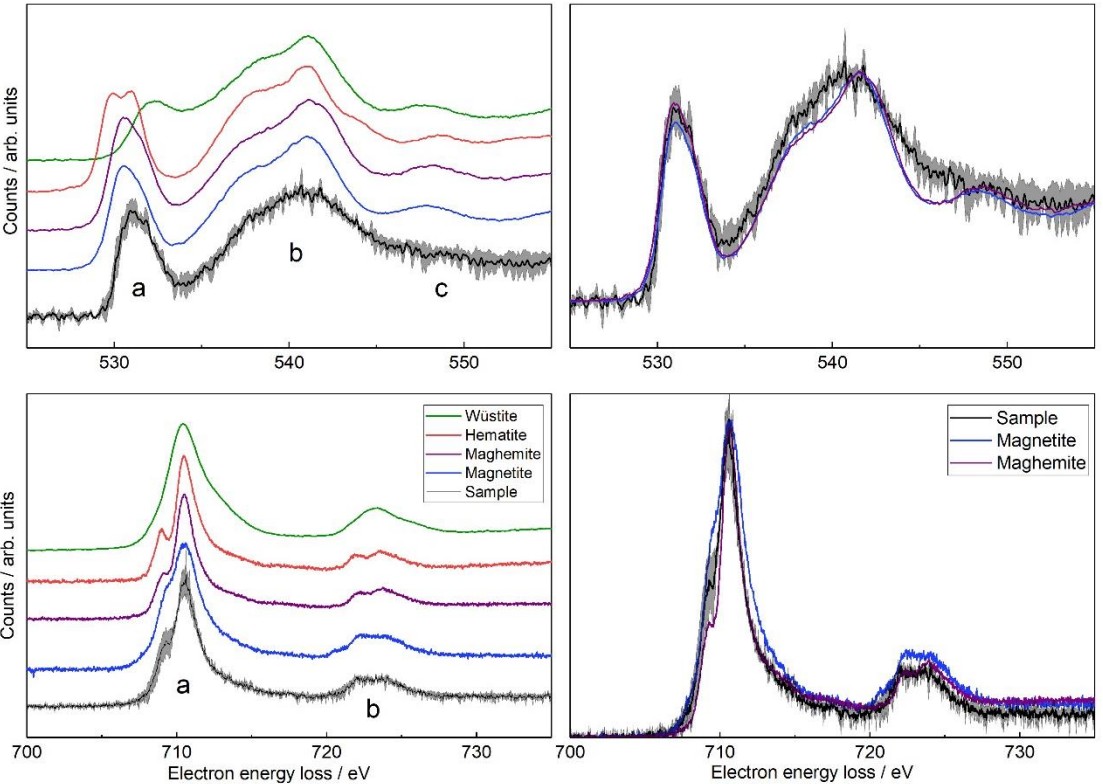

Figure 3. Electron energy loss spectra measured with the TEM compared to spectra for iron oxide standards (Brown et al., 2017; Brown et al., 2001). Top panels: O K-edge. Bottom panels: Fe L edge. Grey shaded regions indicate the experimental uncertainty. The left-hand panels show the spectra offset for clarity, and the right-hand panels show the same spectra (for the sample, magnetite and maghemite) superimposed.

The defined peak *c* is not observed in the sample O K-edge, which we speculate may be due to the poor crystallinity of the particles. This is confirmed by the electron diffraction pattern (Figure S3, supplementary information) where two broad rings are observed coinciding with the intense reflections of magnetite/maghemite. The interplanar distances measured, accounting for a camera calibration error of ± 6 %, were 2.60 Å and 1.47 Å near to the 311 and 440 diffraction planes, respectively (cf. measured distances of 2.57 Å and 1.52 Å in a maghemite standard). A faint ring was observed at an interplanar distance of 2.09 Å, close to the 400 diffraction plane at 2.11 Å. The diffraction pattern cannot distinguish between maghemite and magnetite, since both are based on a spinel crystal structure. However, this analysis does corroborate that the sample composition is very similar to one or other of these minerals. Elemental quantification using EELS resulted in a Fe/O ratio of $0.48 \pm 0.12$ – more oxygen rich than any of the possible compositions hematite/maghemite, magnetite or wüstite (the oxides have Fe/O ratios of 0.67, 0.75 and 1, respectively, i.e. Fe:O = 2:3, 3:4 and 1:1). As such, the composition is most likely to be maghemite-

like, although potentially with additional oxidation or oxygen contamination. This additional oxygen could be another reason for the differences observed in the O K-edge.

The Energy Dispersive X-ray (EDX) spectrum (Figure S4, supplementary information) confirms the presence of Fe and O, though some differences are observed in the intensities of the peaks in the spectra for the agglomerate and the maghemite standard. In the agglomerate spectrum, the intensity of the low energy Fe L-peak is higher than for the standard, which we speculate may be due to fluorescence from excited Cu X-rays from a grid bar, consistent with a relatively large amount of Cu in the agglomerate spectrum. The sample is also more oxygen-rich than the standard. This could result from contamination which was introduced after deposition, prior to TEM imaging. A significant carbon peak is observed in the agglomerate spectrum, suggesting a possible C and O rich hydrocarbon source for this contamination. Alternatively, oxygen could have been introduced within the flow apparatus, by coordination to, or reaction with, an oxygen-rich species $O_3$, forming an oxide coating.

$FeO_3$ is thought to form from the sequential oxidation of Fe by $O_3$ (Fe $\rightarrow$ FeO $\rightarrow$ FeO$_2$ $\rightarrow$ FeO$_3$); the rate coefficients for these three reaction have been measured in the gas phase to be fast (Self and Plane, 2003). The formation of $Fe_2O_3$ smoke analogues in the PAFS apparatus has been previously proposed to occur by polymerization and subsequent re-ordering of $FeO_3$ in the solid phase (Saunders and Plane, 2006); it may be that incomplete re-structuring of the $FeO_3$ has occurred, thus causing the decreased Fe/O ratio. Previous work using the PAFS under comparable experimental conditions obtained a Fe/O ratio of $0.65 \pm 0.06$ (Saunders and Plane, 2006). Although this was suggested to imply the formation of hematite, it would also be consistent with maghemite. Navrotsky et al. (2008) show that for nanoparticles less than ~16 nm in size, maghemite is more stable than hematite since it has a lower surface enthalpy. We therefore conclude that a maghemite-like composition is most likely for the smoke analogues generated using the PAFS.

**3.2 PAFS**

The reduction in intensity of a beam of light from $I_0$ to $I$ as it traverses a distance $l$ through an absorbing medium can be expressed as an optical density (OD) using the Beer-Lambert equation:

E3
$$OD = ln\left(\frac{I_0}{I}\right) = \alpha_{ext} \cdot l$$

where the extinction coefficient $\alpha_{ext}$ arises from both absorption and scattering. The intensity ($I$) at time $t$ is given by the sample spectrum recorded with the particle flow directed through the absorption cell ($I_{n,sa}(\lambda, t)$). A straight line reference fitted to the background spectrum ($I_{n,bg}(\lambda, t)$) yields $I_0$ at time $t$, enabling the time and wavelength-dependent OD to be extracted from the raw spectra. Once the particle size distribution exiting the absorption cell of the PAFS had stabilized, spectra were averaged to obtain one OD spectrum for the iron oxide nanoparticles (Figure 4). As shown by the black shaded area in the bottom panel of Figure 4, the uncertainty in the OD increased significantly at small wavelengths as a result of the decreasing intensity of the spectroscopic lamp and the fall-off in quantum efficiency of the CCD detector. Consequently, the optical data below 350 nm was discarded. At long wavelengths, data above 550 nm was also discarded because the OD decreased below the detection limit. The OD spectrum was also corrected for contributions from the residual precursors used to make the particles. The OD of residual $O_3$ was negligible over the wavelength range of usable experimental data ($\lambda > 350$ nm).

However, the residual $Fe(CO)_5$ spectrum did need to be subtracted. Inspection of the literature
did not yield appropriate data, so the $Fe(CO)_5$ absorption cross section was measured (Figure 6,
top panel). The cross section data is listed in Table S3. The extent of $Fe(CO)_5$ photolysis in the
photolysis cell (Figure 1) was calculated using the flow rates, the lamp irradiance and the
wavelength-dependent absorption cross sections.

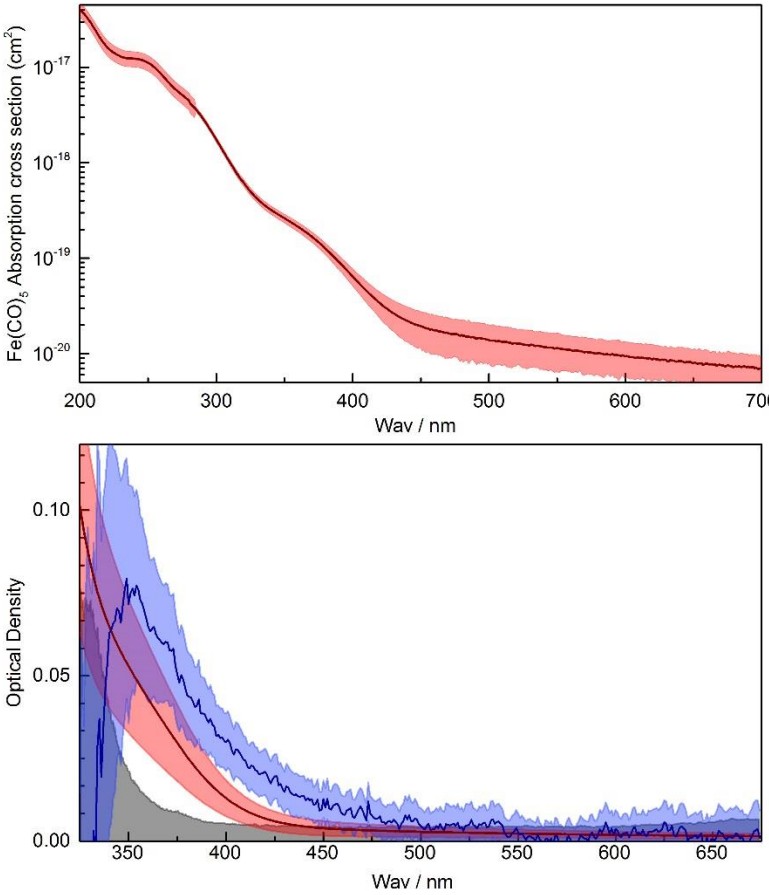


Figure 4. Top panel: Measured $Fe(CO)_5$ absorption cross section ($cm^2$) with experimental uncertainty
indicated by red shading. Bottom panel: Iron oxide particle extinction with the precursor spectrum removed
(blue line) and experimental uncertainty indicated by light blue shading. Also shown is the spectrum for
the $Fe(CO)_5$ present in the absorption cell (red line), with the experimental uncertainty indicated by red
shading. The detection limit for the experiment is shown with the black line and shaded region. Note the
different wavelength ranges in each panel.

The size distribution of agglomerates measured with the SMPS follows an approximate
lognormal distribution peaking around 100 nm radius (demonstrated by the lognormal fit in
Figure 5). A small additional mode is present in the distribution with a peak of approximately 30
nm. The measured size distribution provides a measure of the mobility radius, which is not
necessarily equivalent to the fractal (outer) radius of amorphous particles – these are typically
sized differently to spherical particles in an SMPS as they experience higher drag compared to a
sphere with the same mass (DeCarlo et al., 2004). As such, it should be noted that it may not be
appropriate to use the measured size distribution to calculate the optical extinction. Indeed, some
very large (~2 μm) particles are observed in the TEM images, though these may have resulted
from further agglomeration during deposition on the collection grid. As shown in Figure 6, using
Mie theory with the experimental size distribution over-predicts the OD by at least an order of
magnitude when using literature RIs for hematite, magnetite and wüstite (Hsu and Matijevic,
1985; Longtin et al., 1988; Querry, 1985; Fontijn et al., 1997; Huffman and Stapp, 1973;
Henning and Mutschke, 1997). No equivalent calculation can be performed for maghemite as the
bulk RIs are not available in the literature.

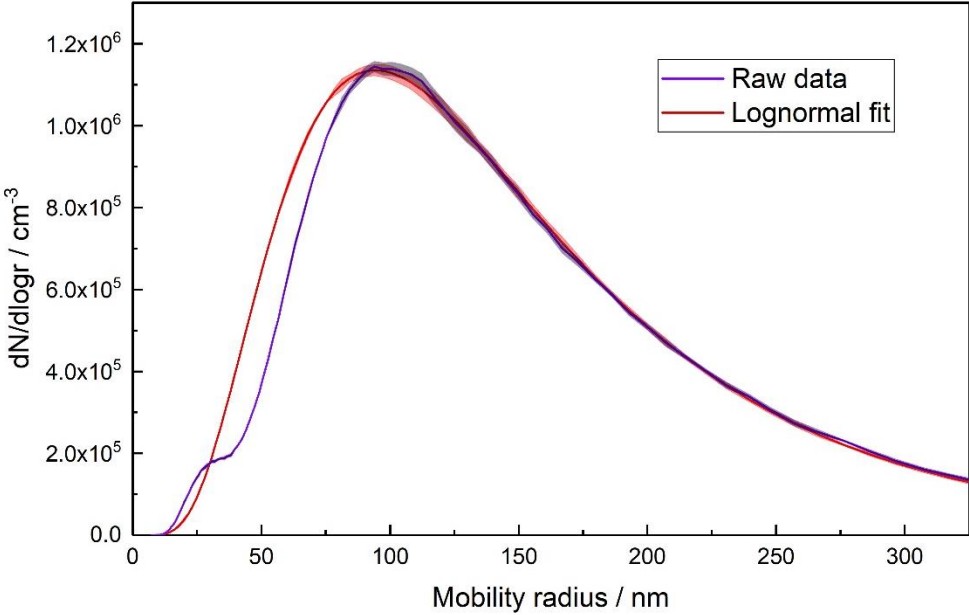


Figure 5. Measured size distribution (purple) and a lognormal fit to the experimental data (red) with shaded
areas indicating the experimental uncertainty.

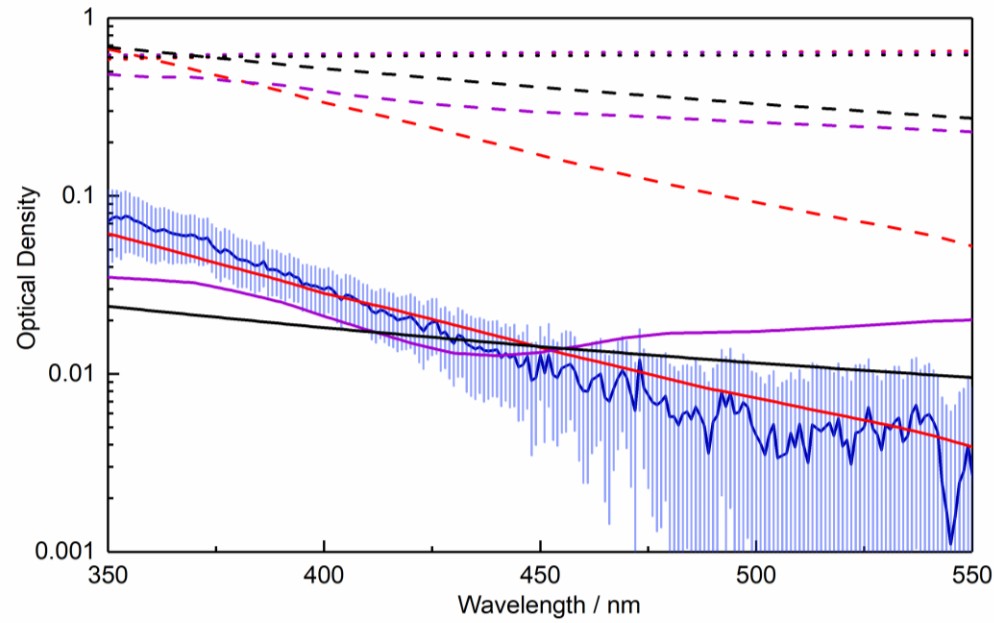


Figure 6. Measured OD as a function of wavelength (blue line), compared with the average OD calculated from literature data for hematite (red, (Hsu and Matijevic, 1985; Longtin et al., 1988; Querry, 1985)), magnetite (purple, (Fontijn et al., 1997; Huffman and Stapp, 1973; Querry, 1985)) and wüstite (black, (Henning and Mutschke, 1997)) using the measured size distribution (dotted lines), the RDG approximation with a monomer concentration derived from the measured size distribution (dashed lines) and the RDG approximation with a monomer concentration fitted to the experimental data (solid lines).

347

An alternative method for calculating the OD of amorphous agglomerates is the Rayleigh-Debye-Gans (RDG) approximation (Sorensen, 2001), where an agglomerate is treated as a monodisperse distribution of primary spheres and the overall agglomerate extinction is calculated by summing those of the individual primary particles. Using a concentration of $r =$ 1.65 nm monomers calculated by integrating the measured size distribution ($4.3 \times 10^{11}$ cm$^{-3}$), Figure 6 shows that the OD is again over-predicted by around an order of magnitude, though a decrease in OD with increasing wavelength is obtained which more closely matches the experimental data. If the monomer concentration is reduced, as would be expected if the observed size distribution is made up of fractal-like particles rather than solid spheres, significantly better fits to the experimental data can be achieved for all species considered, with the best agreement achieved when using hematite RIs (Figure 6). Assuming the RDG approximation holds, a comparison can be made with absorbance data for maghemite (Jain et al., 2009; Tang et al., 2003): for particles on the order of a few nanometers in size, absorption dominates over scattering (for the iron oxides, scattering < 0.01 % absorption). As such, the contribution from scattering to the OD can be neglected, and the absorbance data available in the literature can be arbitrarily scaled for comparison with the experimental data, since scaling the absorbance is equivalent to changing the concentration of monomers in the RDG approximation. This comparison is shown in Figure 7, where the literature data agrees reasonably well with the measured OD. As the literature optical data for hematite and maghemite best replicate the

measured OD, this again suggests the most likely composition to be maghemite-like, given that
the EELS analysis definitively excludes a hematite composition.

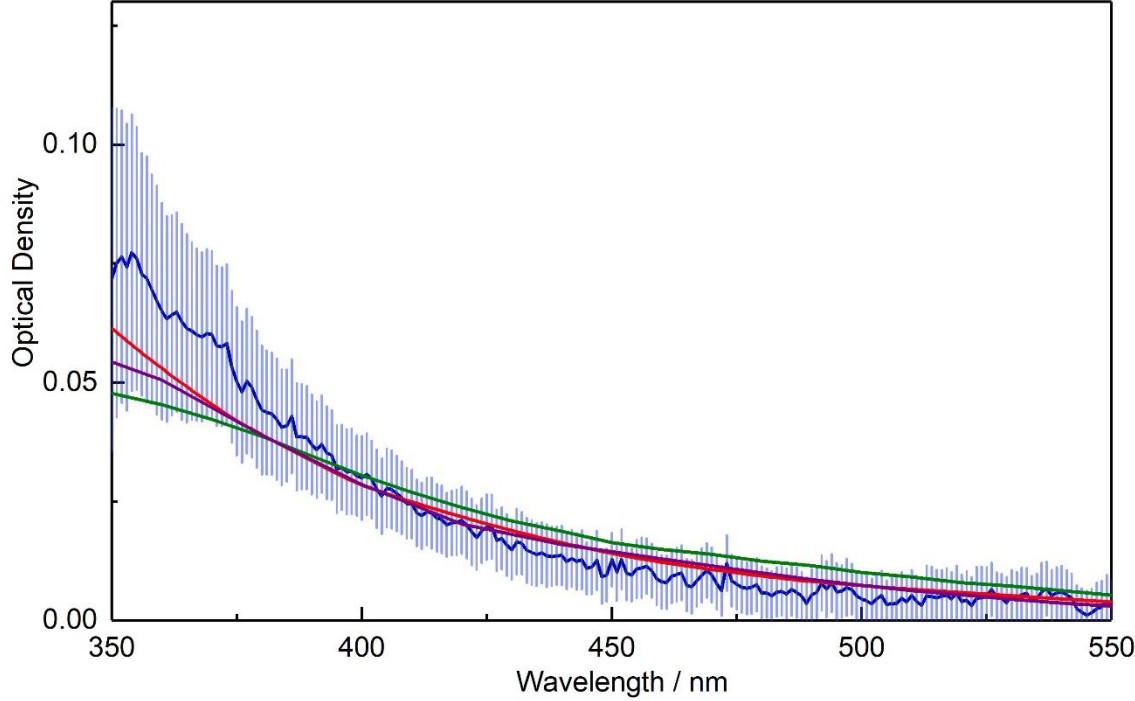



Figure 7. Measured OD (blue line), scaled maghemite OD from Jain et al. (2009) (green line) and scaled
maghemite OD from Tang et al. (2003) (purple line), as a function of wavelength. Also shown is the average
OD calculated from literature data for hematite (red, (Hsu and Matijevic, 1985; Longtin et al., 1988; Querry,
1985)) using the RDG approximation with a monomer concentration fitted to the experimental data (as
shown in Figure 6).

### 3.3 Photochemical Modelling

The previously measured size-dependent absorption efficiencies from the MICE/TRAPS
experiment that were used to derive complex RIs are shown in Figure 8. The solid lines represent
an average of the absorption efficiencies calculated with Mie theory from hematite RIs available
in the literature (Querry, 1985; Bedidi and Cervelle, 1993; Hsu and Matijevic, 1985; Longtin et
al., 1988). The experimental data generally agrees with that from the literature, given the
experimental uncertainties and the variation in literature values. The best agreement is seen at
488 nm, where there is approximately a 20 % difference between the size-dependent literature
average and the experimental values. The absorption efficiencies at 405 nm are around 45 %
larger than the average literature values, whilst still being within the spread of the experimental
error. Those at 660 nm are around 45 % smaller than the average literature values, on the edge of
the range spanned by the experimental errors.

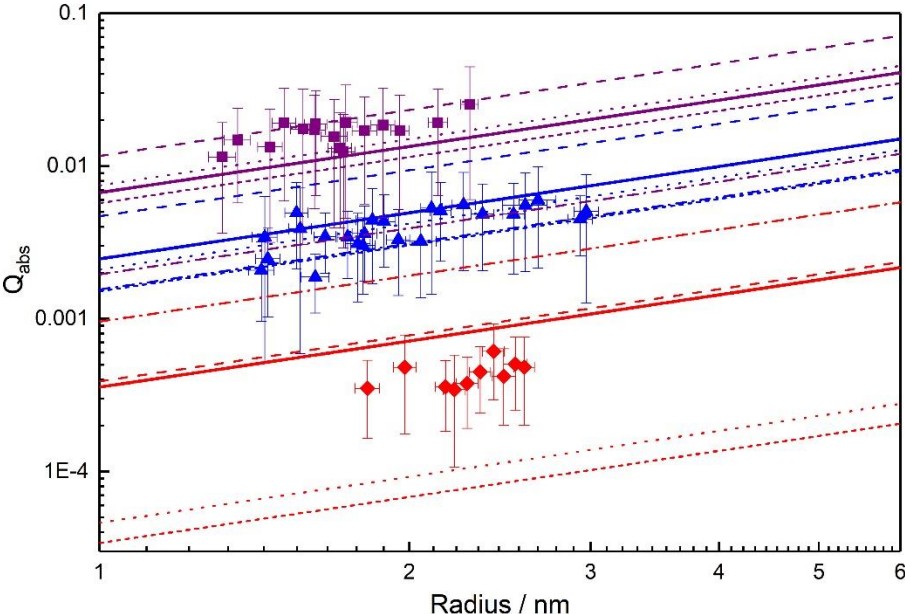


Figure 8. Absorption efficiencies for a range of particle sizes at three different wavelengths: 405 nm
(purple), 488 nm (blue) and 660 nm (red). Also shown are the literature data for hematite particles from
Querry (long dash), Hsu and Matijevic (short dash), Bedidi and Cervelle (dot dash), Longtin et al. (dotted)
and the average (bold lines).

In order to model the PAFS data, at each of the three wavelengths studied with the
MICE/TRAPS (405, 488 and 660 nm), a range of best-fit complex RIs (equation E4, where $n$ and
$k$ are the real and imaginary parts, respectively) was established by iterating over a range of
possible RIs and calculating absorption cross sections for each real-imaginary pair using Mie
theory. The indices resulting in the smallest normalized square difference ($d_{abs,\lambda}$, E5) between the
measured ($C_{abs,exp}$) and calculated ($C_{abs,calc}$) absorption cross section gave the best-fit RIs to the
absorption data at that wavelength (Figure 9).
E4 $$\underline{n} = n + ik$$
E5 $$d_{abs,\lambda} = ((C_{abs,exp} - C_{abs,calc})/C_{abs,exp})^2$$
By neglecting the scattering component in the absorption-dominated OD, the extinction cross
sections measured in the PAFS could be approximated using the absorption cross sections
measured in the MICE/TRAPS. This enabled a best-fit primary particle concentration to be
determined for the PAFS particles, using data from the two wavelengths at which the measured
extinction was above the detection limit (405 nm and 488 nm). Using the best-fit complex RIs
from the MICE/TRAPS data, the PAFS extinction was calculated using the RDG approximation
for a range of primary particle concentrations. At each wavelength (405 or 488 nm), the
normalized square difference between the measured and calculated extinction cross sections was
calculated for each concentration ($\delta_{ext,\lambda}$, equation E6, where $C_{ext,exp}$ and $C_{ext,calc}$ are the
experimental and calculated extinction cross sections, respectively). The $\delta_{ext,\lambda}$ values for the two
wavelengths were summed to derive $\chi^2_{ext}$ (equation E7) and the concentration resulting in the
smallest $\chi^2_{ext}$ value gave the best-fit primary particle concentration, generating the best match to
the measured extinction over the two wavelengths.
Using this best-fit primary particle concentration of $3.14 \times 10^{10}$ cm$^{-3}$, $\delta_{ext,\lambda}$ was calculated for a
range of complex RIs at 405 and 488 nm, with the indices giving the smallest $\delta_{ext,\lambda}$ value
defining the best-fit to the extinction data at each wavelength (Figure 9). The final best-fit RIs at
each wavelength, fitting both the absorption and extinction data, were those that generated the
minimum combined $\delta$ value ($\delta_\lambda$, equation E8). At 660 nm, the final best-fit RIs used were those
which best fit the absorption (gave the minimum $\delta_{abs,\lambda}$).
E6                     $$\delta_{ext,\lambda} = ((C_{ext,exp} - C_{ext,calc})/C_{ext,exp})^2$$
E7                     $$\chi^2_{ext} = \sum \delta_{ext,\lambda}$$
E8                     $$\delta_\lambda = (\delta d_{ext,\lambda} - \delta_{abs,\lambda})^2$$

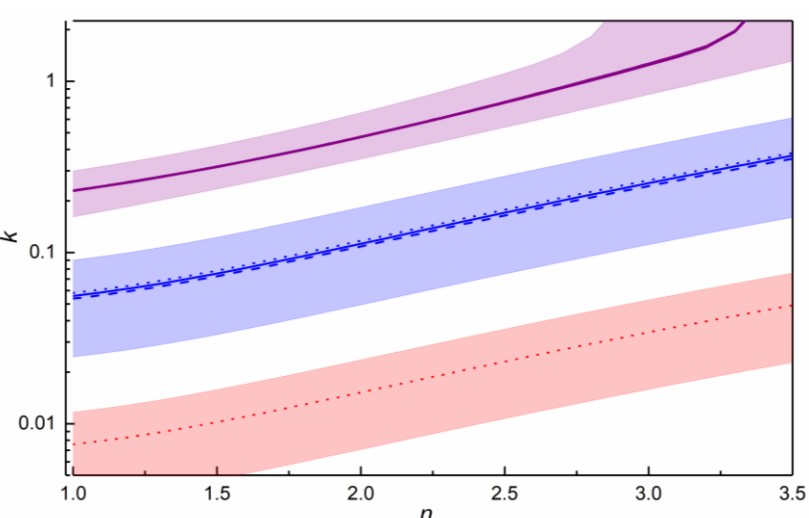

Figure 9. Best-fit RIs $k$ and $n$ for data at 405 nm (purple), 488 nm (blue) and 660 nm (red), for  absorption
(dotted lines), extinction (dashed lines) and the combination (solid lines). Shaded regions indicate where
the resulting absorption and extinction cross sections are within experimental error for both experiments.
As it is possible to reproduce the measured absorption and extinction data at each wavelength
using multiple different combinations of RIs, it is not possible to identify a unique solution for
the wavelength dependence of these parameters. However, one way forward is first to select the
wavelength dependence of the real RIs, which have a much smaller impact on the extinction
cross section than the imaginary RIs in the absorption-dominated regime. The literature data for
hematite from Hsu and Matijevic (1985), Longtin et al. (1988) and Querry (1985) very
satisfactorily fit the experimental data across the whole wavelength range within experimental
errors using the RDG approximation (Figure 6, red line). Thus, for the unidentified particles an
average of the real RIs from these data was used (Figure 10, top panel). Using these real RIs at
405, 488 and 660 nm, the imaginary RI at each wavelength was selected from the best-fit data
(Table S1). The wavelength dependence was then determined by fitting an exponential decay
function through the three values (Figure 10, top panel, Table S1). The wavelength dependent
optical densities calculated using these RIs are shown in Figure 10 (bottom panel). At 405, 488
and 660 nm the calculated absorption efficiencies for a 1.65 nm particle are $1.59 \times 10^{-2}$, $3.19 \times$
$10^{-3}$ and $3.19 \times 10^{-4}$ respectively, compared to the experimentally determined values of $(1.60 \pm$
$1.15) \times 10^{-2}$, $(3.31 \pm 1.92) \times 10^{-3}$ and $(3.19 \pm 1.73) \times 10^{-4}$.

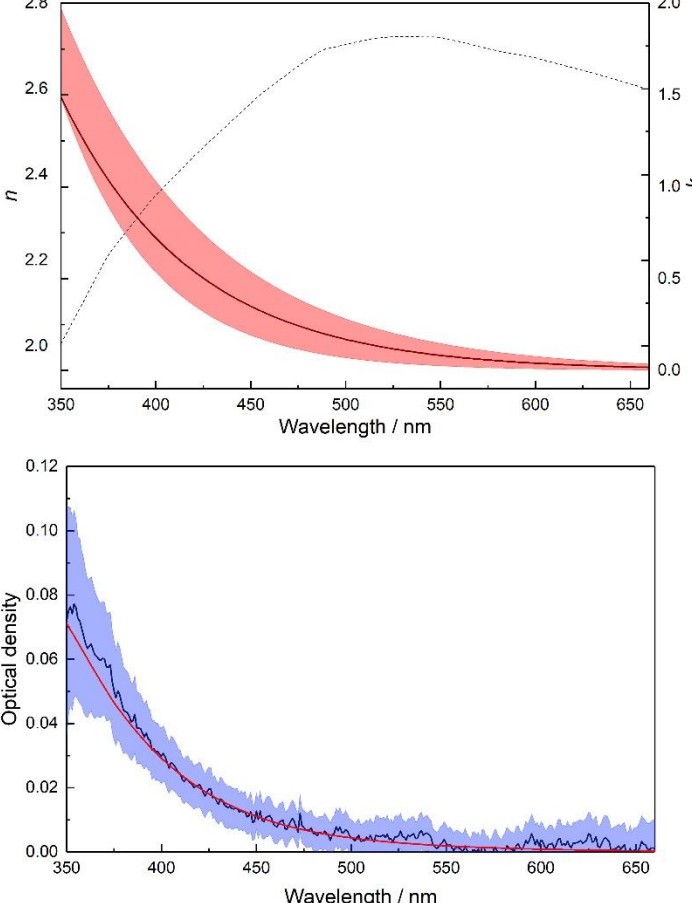

Figure 10. Top panel: Real ($n$) and imaginary ($k$) RIs for maghemite particles (dashed and solid lines,
respectively) with the uncertainty in $k$ indicated with red shading. Bottom panel: Experimental OD (blue)
and the calculated OD (red) using the wavelength dependent RIs for maghemite particles.

**4. Conclusions**
Wavelength-dependent complex RIs have been derived for iron oxide meteoric smoke analogues
generated under atmospherically relevant conditions using two different experimental systems.
Analysis of particles collected from both experiments suggested a maghemite-like composition
to be most likely, although, for the particles produced in the PAFS, a magnetite-like composition
could not be definitively ruled out. Assuming the PAFS particles were indeed maghemite-like,
data from the two experiments was combined using an iteration procedure to determine 'best-fit'
complex RIs that replicate both experimental datasets at 405 nm and 488 nm, and the absorption
data at 660 nm. Values for the real RIs from the literature that generated the closest match to the
measured extinction data (using the RDG approximation for 1.65 nm particles) were used with
the best-fit data to determine the imaginary RIs at wavelengths between 350 and 660 nm.

Despite a number of iron oxides being considered as some of the most probable constituents of
meteoric smoke, maghemite particles have not previously been investigated due to a lack of RIs
available in the literature. Note that the production of maghemite-like particles in the laboratory
using very different experimental conditions demonstrates the potential importance of this species
in the atmosphere. Mesospheric metal chemistry leads to the formation of gas-phase precursors to
MSPs such as iron oxides and hydroxides. The particle production method used in the PAFS
mimics this: UV photolysis of $Fe(CO)_5$ leads to the formation of gas-phase Fe, which reacts with
$O_3$ present in the system to form oxides such as FeO, $FeO_2$ and $FeO_3$. Particles were then allowed
to freely agglomerate in the presence of $O_2$ and $O_3$ – as they would in the atmosphere. Since the
two experiments use a different iron precursors ($Fe(CO)_5$ and $Fe(C_2H_5)_2$), the choice of precursor
does not appear to be a significant factor affecting the composition of particles formed.  The PAFS
operates at standard atmospheric pressure, and the MICE-TRAPS particles are produced at a much
lower pressure of ~60 mbar. Although still higher than in the upper mesosphere, the formation of
similar particles in the two experiments suggests that pressure does not significantly change the
particle properties. Lastly, in the PAFS experiments the $O_3$:$O_2$ ratio used is ~$10^3 \times$ higher than in
the atmosphere. However, the particles in the MICE-TRAPS apparatus are produced in the
presence of $O_2$ only, and still form maghemite-like particles. As mentioned in the Introduction, the
most likely candidates for smoke particles are iron oxides and silicates, but it is not known whether
these occur in a single phase or separate distinct phases. For this reason, there is a need for further
studies on the optical properties of maghemite.
The present study also demonstrates that the RDG approximation is more appropriate than Mie
theory to model the optical properties of fractal-like MSPs, since Mie theory over-predicts the
optical extinction by at least an order of magnitude across the wavelength range studied. This
supports the earlier work of Saunders et al. (2007) and is important since current studies with the
SOFIE satellite calculate MSP extinction using Mie theory for a distribution of spherical
particles (Hervig et al., 2017). Nevertheless, the fact that the derived complex RIs generated
good fits to both the absorption and extinction produced by crystalline and amorphous particles
in the MICE-TRAPS and PAFS experiments, respectively, lends confidence to the idea that it is
appropriate to use the RIs for bulk (crystalline) species to represent amorphous MSPs for the
purposes of their characterisation.
Though the complex RIs derived for the particles do not represent a unique solution to the
wavelength dependence across the wavelength range studied, they provide good fits to both the
experimental extinction and absorption in the two experiments. As such, since the important
parameter for MSP characterization in the atmosphere is the particle extinction, these RIs should
be applicable across this wavelength range (using different combinations of best-fit RIs incurs an
error of < 0.4 % in the particle extinction at 405, 488 and 660 nm). However, in order for these
RIs to be used with data from the SOFIE satellite, the wavelength range would need to be extended
further into both ultra-violet and infra-red wavelengths. With the current data, although
extrapolation to a wavelength of 330 nm may be feasible, it is not possible to extrapolate to the
other wavelengths currently used for SOFIE analysis (867 and 1037 nm); the difference in
wavelength is too great, given the unpredictable variation in RIs usually observed across wide
wavelength ranges. Nevertheless, the RIs could be used in global climate models to probe the
optical properties of meteoric smoke and make comparisons to observations.

**Data availability**
The refractive index data and $Fe(CO)_5$ absorption spectrum are archived at the Leeds University
PETAL (PetaByte Environmental Tape Archive and Library;
http://www.see.leeds.ac.uk/business-and-consultation/facilities/petabyte-environmental-tape-
archiveand-library-petal/) and are available upon request to JMCP.
**Author contribution**
The PAFS experiments were designed by TA, AJ and JP, and carried out by TA, who also
performed the data analysis. The photochemical model was designed and written by TA, based
on code written by JB. The MICE/TRAPS experiments were designed by MN, DD and TL.
Experiments were carried out by MN and TA. MN performed the data analysis. TA prepared the
manuscript with contributions from all co-authors. DD, JP and TL supervised the project.
**Competing interests**
There are no competing interests.
**Acknowledgements**
This work was supported by the UK National Environment Research Council (NERC). TA has a
research studentship funded by the NERC SPHERES doctoral training program, which included
funding for a research placement at the Karlsruhe Institute of Technology (KIT), Institute of
Meteorology and Climate Research. We would also like to thank Andy Brown at the Leeds
Electron Microscopy and Spectroscopy Centre (LEMAS) for his invaluable help with the TEM,
EDX and EELS analysis, and for providing data for the iron oxide standards. We thank Prof.
Dwayne Heard for the loan of the SMPS instrument.

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
