# Peer review of "Optical properties of meteoric smoke analogues"

_Atmospheric Chemistry and Physics, 2019_

## Referee Comment (RC1) · Anonymous Referee #1 · 8 Jul 2019

This manuscript reports the combination of two experiments in order to quantify complex refractive indices (i.e., real and imaginary parts) of maghemite (gamma-Fe2O3). This is an important data set since iron oxides are among the candidates for meteoric smoke particles which in turn have been proposed to be involved in a wide variety of atmospheric phenomena.

In order to derive both the real and imaginary parts of refractive indices, two data sets were analyzed: one from a photochemical aerosol flow reactor in which the extinction (scattering plus absorption) of iron oxide analogues was measured in the wavelength range 325-675 nm. These data were then combined with maghemite absorption coefficients measured in an independent experimental system.

Overall the manuscript is very well written and certainly warrants publication provided that the following minor comments are taken care of:

[Figure]

1) There is only one critical point that I see and that is that in the two experiments the particles were generated using completely different experimental techniques. Arguments are presented that allow to identify maghemite as the likely composition of particles in both experimental setups. However, in particular with regard to Figure 3, I am wondering if the authors could try to be a bit more compelling that the particles are indeed made of maghemite. In particular, I would like to see a direct comparison of the measured spectra with the shown spectra for iron oxide standards. While I do agree that some features of maghemite fit the measured spectra well, others do not appear to fit perfectly. For example in the Fe L edge region at 720-725nm the reference spectrum for maghemite shows a clear double peak which I can't see in the measurements. I recommend to plot one spectrum on top of the other and be a bit more critical in the attribution. Also a cautionary note with regard to the proper identification should be added to the conclusions and abstract.

2) Line 204: Section 3.2 should be 3.1

3) Figure 4: The two panels have different wavelength-ranges which confused me initially. This should at least be indicated in the figure caption.

4) Line 309: Please provide a reference for the statement that the mobility radius is an upper limit to the fractal radius.

5) I might have missed it: in Figure 5, radii from 30-100nm, there is a clear mismatch between the measured size distribution and the lognormal fit. This should at least be stated and possibly discussed in the text.

6) Figure 10 and lines 415-418: In the text the authors write that the literature data is for hematite and not maghemite. Subsequently they use the real refractive indices for hematite. Is this really consistent? I admit to be confused. Please explain.

7) Line 443-445: The authors argue that since maghemite emerged as the dominant species in the laboratory experiments it might also play a role in the atmosphere. Can

this inference really be drawn? How comparable are the conditions in the laboratory and the atmosphere?

---

## Referee Comment (RC2) · Anonymous Referee #2 · 14 Jul 2019

The paper by Aylett et al presents a laboratory study determining optical characteristics of analogs of meteoric smoke particles. Specifically, refractive indices for MSP analogs concluded to be maghemite are found. MSPs play a significant role in the atmosphere and so this is a worthwhile study.

The paper is well written with clear, succinct, and informative text. The analysis is well described. The methodology and logic are clear. My comments are only minor, but include one discussion that the authors should consider. Overall this is a high-quality paper that that is very appropriate for publication in ACP.

A significant motivation for this work is the SOFIE results. The authors state that those results are questionable due to the SOFIE-analysis assumption that MSPs are essentially crystalline i.e. refractive indices measured from crystal forms of possible MSP components can be utilized to infer MSP composition. In this study, the MSP analogs

were found to be amorphous and not crystalline. Does this result further call into question the SOFIE results? The authors were not able to determine optical properties at the wavelengths SOFIE utilized and so formed no conclusions regarding the SOFIE work. But couldn't the authors make the crystalline assumption as done with SOFIE to see if that resulted in large errors in the refractive indices they determined? And if done, would this impact the interpretation of the SOFIE data? More discussion with regard to the existing remote sensing results would strengthen this paper.

Other minor comments:

There are numerous places where space should be, but were not: For examples Line 173-174; 326-327

There is no Section 3.1

---

## Author Comment (AC1) · 11 Aug 2019

We thank both reviewers for their very helpful comments on the paper, which we have addressed in a revised version of the manuscript.

Response to reviewer 1

Comment 1: There is only one critical point that I see and that is that in the two experiments the particles were generated using completely different experimental techniques. Arguments are presented that allow to identify maghemite as the likely composition of particles in both experimental setups. However, in particular with regard to Figure 3, I am wondering if the authors could try to be a bit more compelling that the particles are indeed made of maghemite. In particular, I would like to see a direct comparison of the measured spectra with the shown spectra for iron oxide standards. While I do agree that some features of maghemite fit the measured spectra well, oth-

ers do not appear to fit perfectly. For example in the Fe L edge region at 720-725nm the reference spectrum for maghemite shows a clear double peak which I can't see in the measurements. I recommend to plot one spectrum on top of the other and be a bit more critical in the attribution. Also a cautionary note with regard to the proper identification should be added to the conclusions and abstract.

Author response: Though there is not a perfect fit between the sample EEL spectra and those for the maghemite standard, the authors consider that the sample spectra most closely resemble those for this standard. As suggested, additional panels have been added to Figure 3 showing the Fe L-edges and I K-edges plotted on top of one another to demonstrate this, and additional discussion of these panels has been added to the text for clarification. We also note that the identification of maghemite as the most likely composition is not solely based upon the EEL spectra, but also upon the Fe:O ratios, optical analysis, and previous experimental data from the PAFS apparatus operating under comparable conditions. A cautionary note has been added to both the conclusions and abstract stating that a magnetite-like composition is also possible. Changes made: Figure 3 – two panels added showing superimposed spectra.

Figure caption changed to: "Figure 3. Electron energy loss spectra measured with the TEM compared to spectra for iron oxide standards (Brown et al., 2017;Brown et al., 2001). Top panels: O K-edge. Bottom panels: Fe L edge. Grey shaded regions indicate the experimental uncertainty. The left-hand panels show the spectra offset for clarity, and the right-hand panels show the same spectra (for the sample, magnetite and maghemite) superimposed."

Changes to the paragraph starting at line 232 (shown in bold type): "In the case of both magnetite and maghemite there are no distinctive features in either the O K or Fe L-edges to easily distinguish between the two species. Nonetheless, upon closer inspection (Figure 3, right-hand panels) the sample spectra more closely resemble those for the maghemite standard compared to those for magnetite. For the Fe L-edge, although the shoulder on the low energy side of peak a is larger in the sample spectrum

than that observed for maghemite, it is more well defined than the shoulder seen in the magnetite spectrum. Furthermore, the profile of peak a more closely follows that for the maghemite standard on both the high and low energy sides. Though a defined double-peak structure is not observed in peak b, both the peak profile on the high energy side and the height of the peak more closely resemble maghemite. For the O K-edge, although there are some differences between the sample spectra and those for both standards (notably the lack of a defined peak c), there are minimal differences between the spectra for the maghemite and magnetite standards. For this reason, though the profile of peak a more closely follows that of maghemite, it is not possible to distinguish between the two species from the O K-edge spectra alone." Sentence added at line 266: "This additional oxygen could be another reason for the differences observed in the O K-edge."

The following change in the Abstract (lines 19/20) for additional clarity: "Analysis using Transmission Electron Microscopy (TEM), Energy Dispersive X-ray spectroscopy (EDX) and Electron Energy Loss Spectroscopy (EELS) suggested the particles were most likely maghemite-like ($\gamma$-Fe2O3) in composition, though a magnetite-like composition could not be completely ruled out. Assuming a maghemite-like composition, the optical extinction coefficients measured using the PAFS were combined with maghemite absorption coefficients measured using a complementary experimental system (the MICE-TRAPS) to derive complex refractive indices that reproduce both the measured absorption and extinction."

Further text has been added to the Conclusions at lines 456-463: "Wavelength-dependent complex RIs have been derived for iron oxide meteoric smoke analogues generated under atmospherically relevant conditions using two different experimental systems. Analysis of particles collected from both experiments suggested a maghemite-like composition to be most likely. Although, for the particles produced in the PAFS, a magnetite-like composition could not be definitively ruled out. Assuming the PAFS particles were indeed maghemite-like, data from the two experiments was

combined using an iteration procedure to determine 'best-fit' complex RIs that replicate both experimental datasets at 405 nm and 488 nm, and the absorption data at 660 nm."

Comment 2: Line 204: Section 3.2 should be 3.1 Author response: This has been corrected, as well as the subsequent section headings.

Comment 3: Figure 4: The two panels have different wavelength-ranges which confused me initially. This should at least be indicated in the figure caption.

Author response: A sentence has been added to clarify this: "Figure 4. Top panel: Measured Fe(CO)5 absorption cross section (cm2) with experimental uncertainty indicated by red shading. Bottom panel: Iron oxide particle extinction with the precursor spectrum removed (blue line) and experimental uncertainty indicated by light blue shading. Also shown is the spectrum for the Fe(CO)5 present in the absorption cell (red line), with the experimental uncertainty indicated by red shading. The detection limit for the experiment is shown with the black line and shaded region. Note the different wavelength ranges in each panel."

Comment 4: Line 309: Please provide a reference for the statement that the mobility radius is an upper limit to the fractal radius.

Author response: we have removed this statement, and changed the text from lines 324-330: "The measured size distribution provides a measure of the mobility radius, which is not necessarily equivalent to the fractal (outer) radius of amorphous particles – these are typically sized differently to spherical particles in an SMPS as they experience higher drag compared to a sphere with the same mass (DeCarlo et al., 2004). As such, it should be noted that it may not be appropriate to use the measured size distribution to calculate the optical extinction. Indeed, some very large ($\sim$2 $\mu$m) particles are observed in the TEM images, though these may have resulted from further agglomeration during deposition on the collection grid."

Comment 5: I might have missed it: in Figure 5, radii from 30-100nm, there is a clear mismatch between the measured size distribution and the lognormal fit. This should at least be stated and possibly discussed in the text.

Author response: The concentration of monomers used for subsequent analysis was derived from the actual measured size distribution, and not from the lognormal fit – this fit was shown just to indicate that the size distribution followed approximately a lognormal distribution. The presence of this additional mode has now been mentioned in the text at lines 321-324: "The size distribution of agglomerates measured with the SMPS follows an approximate lognormal distribution peaking around 100 nm radius (demonstrated by the lognormal fit in Figure 5). A small additional mode is present in the distribution with a peak of approximately 30 nm."

Comment 6: Figure 10 and lines 415-418: In the text the authors write that the literature data is for hematite and not maghemite. Subsequently they use the real refractive indices for hematite. Is this really consistent? I admit to be confused. Please explain.

Author response: A range of best-fit RIs have been derived for the meteoric smoke analogues at each of the three wavelengths, 405 nm, 488 nm and 660 nm (Figure 9). As stated in the text, it is not possible to identify a unique solution for the wavelength dependence of the RIs across this wavelength range, given that any combination of the best-fit RIs (at 405 nm, 488 nm and 660 nm) would provide a good match to the experimental data (both absorption and extinction).

The overall particle extinction is largely controlled by the wavelength-dependence of the imaginary RI, which follows an approximately exponential-type decay in order to fit the experimental data. However, the wavelength-dependence of the real RI is unknown. As such, the real RI for hematite was used in place of arbitrarily defining a wavelength-dependence for this parameter. The best-fit RIs calculated for the particles were then used to obtain the imaginary part of the RI from the real RI at each of these wavelengths. An exponential fit through these imaginary RIs produced the

wavelength-dependence of the imaginary RI.

The literature RI data for hematite was used since: 1. The hematite RI data provides the best fit to the experimental extinction when using the RDG approximation with a fitted monomer concentration (Figure 6). 2. The extinction calculated using this RI data also closely matches the scaled maghemite absorbance data from the literature (Figure 7). 3. The real RIs of all the iron oxide particles available in the literature are broadly similar, showing an increase with increasing wavelength across the wavelength range studied (between $\sim 2.0 - 2.7$). In light of this, one possible solution to the wavelength dependence of the complex RIs was derived using the literature real RIs for hematite. The RIs derived in this manner are not intended to be a unique solution to the wavelength-dependence of the RIs across the wavelength range studied, but to provide one possible solution that is able to reproduce the data from both experiments. Though it is not a unique solution, the important parameter for MSP characterisation in the atmosphere is the particle extinction. As such, since the derived RIs satisfactorily reproduce the measured extinction, the derived RIs should still be functional for this purpose over the wavelength range studied.

Changes made: The OD calculated using the RDG approximation (with a reduced monomer concentration) for Hematite (as shown in Figure 6) has also been added to Figure 7, to show the similarity of the spectra for Hematite and Maghemite more clearly.

The figure caption is revised accordingly: "Figure 7. Measured OD (blue line), scaled maghemite OD from Jain et al. (2009) (green line) and scaled maghemite OD from Tang et al. (2003) (purple line), as a function of wavelength. Also shown is the average OD calculated from literature data for hematite (red, (Hsu and Matijevic, 1985;Longtin et al., 1988;Querry, 1985)) using the RDG approximation with a monomer concentration fitted to the experimental data (as shown in Figure 6)."

The Conclusions are updated to clarify the points discussed above (line 494-506): "Though the complex RIs derived for the particles do not represent a unique solution to the wavelength dependence across the wavelength range studied, they provide good fits to both the experimental extinction and absorption in the two experiments. As such, since the important parameter for MSP characterization in the atmosphere is the particle extinction, these RIs should be applicable across this wavelength range (using different combinations of best-fit RIs incurs an error of < 0.4 % in the particle extinction at 405, 488 and 660 nm). However, in order for these RIs to be used with data from the SOFIE satellite, the wavelength range would need to be extended further into both ultra-violet and infra-red wavelengths. With the current data, although extrapolation to a wavelength of 330 nm may be feasible, it is not possible to extrapolate to the other wavelengths currently used for SOFIE analysis (867 and 1037 nm); the difference in wavelength is too great, given the unpredictable variation in RIs usually observed across wide wavelength ranges. Nevertheless, the RIs could be used in global climate models to probe the optical properties of meteoric smoke and make comparisons to observations."

Comment 7: Line 443-445: The authors argue that since maghemite emerged as the dominant species in the laboratory experiments it might also play a role in the atmosphere. Can this inference really be drawn? How comparable are the conditions in the laboratory and the atmosphere?

Author response: Since the production mechanisms are very different in both experiments, yet in both cases the particle composition was found to be most similar to maghemite, we suggest that this iron oxide conformer may potentially be important in the atmosphere – though we stress the need for further studies on such iron oxide particles in this context before any conclusions can be drawn. This is now stated at line 466:

"Note that the production of maghemite-like particles in the laboratory using very different experimental conditions demonstrates the potential importance of this species in the atmosphere. Mesospheric metal chemistry leads to the formation of gas-phase precursors to MSPs such as iron oxides and hydroxides. The particle production method

used in the PAFS mimics this: UV photolysis of Fe(CO)5 leads to the formation of gas-phase Fe, which reacts with O3 present in the system to form oxides such as FeO, FeO2 and FeO3. Particles were then allowed to freely agglomerate in the presence of O2 and O3 – as they would in the atmosphere. Since the two experiments use a different iron precursors (Fe(CO)5 and Fe(C2H5)2), the choice of precursor does not appear to be a significant factor affecting the composition of particles formed. The PAFS operates at standard atmospheric pressure, and the MICE-TRAPS particles are produced at a much lower pressure of ∼60 mbar. Although still higher than in the upper mesosphere, the formation of similar particles in the two experiments suggests that pressure does not significantly change the particle properties. Lastly, in the PAFS experiments the O3:O2 ratio used is ∼103 ïĆť higher than in the atmosphere. However, the particles in the MICE-TRAPS apparatus are produced in the presence of O2 only, and still form maghemite-like particles. As mentioned in the Introduction, the most likely candidates for smoke particles are iron oxides and silicates, but it is not known whether these occur in a single phase or separate distinct phases. For this reason, there is a need for further studies on the optical properties of maghemite."

Response to reviewer 2

Comment 1: A significant motivation for this work is the SOFIE results. The authors state that those results are questionable due to the SOFIE-analysis assumption that MSPs are essentially crystalline i.e. refractive indices measured from crystal forms of possible MSP components can be utilized to infer MSP composition. In this study, the MSP analogs were found to be amorphous and not crystalline. Does this result further call into question the SOFIE results? The authors were not able to determine optical properties at the wavelengths SOFIE utilized and so formed no conclusions regarding the SOFIE work. But couldn't the authors make the crystalline assumption as done with SOFIE to see if that resulted in large errors in the refractive indices they determined? And if done, would this impact the interpretation of the SOFIE data? More discussion with regard to the existing remote sensing results would strengthen this paper.

Author response: In this study, the MSP analogues produced with the PAFS were indeed found to be amorphous and not crystalline. However, the particles produced with the MICE/TRAPS were crystalline. In modelling the optical properties of the MSP analogues, this study inherently assumes that the RIs from crystalline particles are applicable to amorphous particles by equating the absorption and extinction cross sections from the two experiments. In making this assumption, and using the RDG approximation to model the extinction of the amorphous particles, wavelength-dependent complex RIs were derived that generated very good fits to both datasets, thus suggesting that this assumption is valid. This is now discussed in the Conclusions at lines 484-493: "The present study also demonstrates that the RDG approximation is more appropriate than Mie theory to model the optical properties of fractal-like MSPs, since Mie theory over-predicts the optical extinction by at least an order of magnitude across the wavelength range studied. This supports the earlier work of Saunders et al. (2007) and is important since current studies with the SOFIE satellite calculate MSP extinction using Mie theory for a distribution of spherical particles (Hervig et al., 2017). Nevertheless, the fact that the derived complex RIs generated good fits to both the absorption and extinction produced by crystalline and amorphous particles in the MICE-TRAPS and PAFS experiments, respectively, lends confidence to the idea that it is appropriate to use the RIs for bulk (crystalline) species to represent amorphous MSPs for the purposes of their characterisation."

Comment 2: There are numerous places where space should be, but were not: For examples Line 173-174; 326-327.

Author response: Spaces added in the text at lines 174-175, 222, 249, 332-334, 342-343, 359-360, 381-382.

Comment 3: 'There is no section 3.1'

Author response: This has been changed in the text, as well as the subsequent section headings.